# Development of Biodegradable Flame-Retardant Bamboo Charcoal Composites, Part I: Thermal and Elemental Analyses

**DOI:** 10.3390/polym12102217

**Published:** 2020-09-27

**Authors:** Shanshan Wang, Liang Zhang, Kate Semple, Min Zhang, Wenbiao Zhang, Chunping Dai

**Affiliations:** 1Department of Engineering, Zhejiang Provincial Collaborative Innovation Center for Bamboo Resources and High-Efficiency Utilization, Zhejiang A&F University, Hangzhou, Zhejiang 311300, China; wangshanshan430@163.com (S.W.); chaselz520103@163.com (L.Z.); zhang888@rish.kyoto-u.ac.jp (M.Z.); 2Department of Wood Science, Faculty of Forestry, University of British Columbia, 2900-2424 Main Mall Vancouver, BC V6T 1Z4, Canada; katherine.semple@ubc.ca

**Keywords:** bamboo charcoal, aluminum hypophosphite, polylactic acid, composites, flame retardancy

## Abstract

In this study, bamboo charcoal (BC) was used as a substitute filler for bamboo powder (BP) in a lignocellulose-plastic composite made from polylactic acid (PLA), with aluminum hypophosphite (AHP) added as a fire retardant. A set of BC/PLA/AHP composites were successfully prepared and tested for flame-retardancy properties. Objectives were to (a) assess compatibility and dispersibility of BC and AHP fillers in PLA matrix, and (b) improve flame-retardant properties of PLA composite. BC reduced flexural properties while co-addition of AHP enhanced bonding between PLA and BC, improving strength and ductility properties. Adding AHP drastically reduced the heat release rate and total heat release of the composites by 72.2% compared with pure PLA. The formation of carbonized surface layers in the BC/PLA/AHP composites effectively improved the fire performance index (FPI) and reduced the fire growth index (FGI). Flame-retardant performance was significantly improved with limiting oxygen index (LOI) of BC/PLA/AHP composite increased to 31 vol%, providing a V-0 rating in UL-94 vertical flame test. Adding AHP promoted earlier initial thermal degradation of the surface of BC/PLA/AHP composites with a carbon residue rate up to 40.3%, providing a protective layer of char. Further raw material and char residue analysis are presented in Part II of this series.

## 1. Introduction

The enormous range and waste volumes of non-biodegradable petroleum-derived plastics produced over past decades are now a leading source of land and ocean pollutants and hazards for marine life. Biodegradable alternatives such as polylactic acid (PLA) have been introduced over the past 40 years in an attempt to reduce the future volume of non-degradable single-use plastic items and packaging [1]. Starches and sugars extracted from industrial crops like corn, sugarcane and sugar beet are fermented to obtain lactide and lactic acid monomers. PLA is now the second most widely used plastic type for food-grade and non-food packaging and implements globally in terms of volume, and the most successfully commercialized type of biodegradable bioplastics. Other more specialized uses include medical meshes and scaffolds, electrical engineering, thermoplastic 3-D printing polymers [2,3]. PLA is compostable (under correct conditions) and can be processed into a wide variety of products on standard plastics processing equipment. However, it loses its thermal molecular weight stability at temperatures above its low melting point, limiting its processing temperatures and residence times [4,5]. Because of the high cost of PLA, its application is limited to high-value thin films, rigid hot forming, cold food and beverage containers and coated paper. However, this is expected to expand to a broader range of packaging products as its processing costs and technologies evolve [6]. Like polystyrene (PS), PLA is also widely used for foamed materials; as pure or in combination with lower cost fillers like starch [7,8].

Whilst non-toxic and safe for food packaging, pure PLA finds limited application in this area and as an engineering plastic due to high cost, low heat distortion temperature, high brittleness and low impact strength, low viscosity, low thermal stability, high moisture sensitivity, medium gas barrier properties, and low solvent resistance (including alcohols and water), limiting its applications for food and liquid packaging without modifications [6,9,10]. It has a very low deformation and distortion threshold as the glass transition point for the crystalline component is just 57 °C, and melting point (MP) at 180 °C; thermal degradation begins at closer to 200 °C [11]. Pure PLA has a very low melt viscosity, undergoing severe melt dripping which exacerbates fire propagation [12].

Simple, low-cost powdered fillers such as eggshells and bamboo charcoal (BC) added in small amounts can also significantly tailor the mechanical, moisture absorption and thermal/combustion properties of PLA films and rigid forms. Bamboo charcoal (BC) has good interfacial properties with hydrophobic plastic polymers, including PLA, as it is much more porous with a higher surface area to weight ratio—250–300 m^2^/g compared with 50 m^2^/g for wood charcoal [13]. Past works incorporating BC into PLA have produced mixed results. Srisuk et al. (2019) [14] found slightly decreased shear viscosity, lower tensile strength and elongation at break of the BC-PLA composites. The optimal BC content in the composites was just 2.82 wt. %, and did not change the hardness compared with neat PLA. Similarly, Zawawi et al. (2019) [15] found significantly reduced tensile strength and Izod impact strength with increasing BC addition up to 20%. Elastic modulus increased significantly (80%) and SEM of composites containing 20 wt. % BC filler showed a brittle fracture surface. The problem was believed to be due to agglomeration of BC particles during mixing. Lau (2014) and Ho et al. (2015) [16,17] successfully combined up to 7.5% BC (10 μm particle size) into PLA, increasing its impact resistance and reducing the effects of water absorption on mechanical properties. PLA with 7.5% BC increased in maximum tensile strength, flexural strength and ductility index (DI) by 43%, 99% and 52%, respectively, compared to pure PLA [17]. To improve the integration and interfacial properties of BC in PLA composites, 10%–15% NaOH solutions were used by Qian et al. (2018) [18] to modify ultra-fine bamboo charcoal (UFBC) to increase its specific surface area, porosity and surface activity. Maximum tensile strength and modulus of the BC-PLA composites were significantly increased to 65.4 MPa and 1 GPa, respectively. Wang et al. (2020) [19] were able to incorporate a high BC content of 25 wt. %, producing PLA composites with tensile strength of 38.1 MPa and flexural strength (MOR) of 45.3 MPa. Adding higher amounts of BC to PLA to produce disposable packaging may also be beneficial for its compostability. 

Flame retardancy improvement of PLA is a high priority for improving the performance and extending the applications of PLA composites. Smoke, toxic gas release and rapid heat release and flame spread from burning polymers are the most significant threats to workers and firefighters [20,21] and large volumes of flammable packaging material can pose a large potential risk. Additives that provide PLA with flame retardancy sometimes cause problems such as loss of mechanical and thermal properties and degradation of the polyester matrix, which can affect processing and end product performance. Much research has been focused on adding a vast array of inorganic fillers and high surface area nano-materials such as nano-silica, nano-clays (e.g., montmorillonite), graphene nanosheets, carbon nano-fibers and tubules, metal oxides, layered double hydroxide (LDA), CaCO_3_, and flame-retardant grafted nanocellulose to PLA and other polymers to improve thermal resistance for processing and flame retardancy and smoke suppression [21,22,23,24]. These can provide good intumescent fire- and smoke-retardant effects, but engineered nano-particles are still relatively costly, often need to be used in uneconomic quantities, can suffer from agglomeration issues during processing, and pose health and environmental contamination concerns [25].

Another possible, cheaper avenue for enhancing the flame retardancy of PLA may be the co-addition of BC powder with lower cost inorganic fire retardants such as aluminum hypophosphite (AHP). AHP is a very effective flame retardant for PLA [26,27]. A UL 94 V-0 flammability rating can be reached for PLA/(AHP-20%) as AHP promotes very effective protective char formation on the surface of PLA. Adding BC filler may effectively further reduce the heat transfer rate in the composite material during the combustion process, enhancing its flame-retardant properties. Li et al. (2014) [24] found adding up to 8% BC to wood plastic composite (WPC) increased the temperature at which 30 and 50% mass loss occurs. Wei et al. (2018) [28] added 5% BC and 10% intumescent flame retardants (IFR) to PLA, giving it a UL-94 V-0 vertical flame test rating. The LOI reached 31.6 vol%, demonstrating enhanced flame-retardant properties of the composites. 

In this further study, combinations of BC and AHP were added to PLA via the conventional melting blending process used for WPCs, with the objective of assessing processability and further improving the flame-retardant properties of PLA. The aim of this study was to measure the effect of different levels of BC and AHP addition on the flame-retardant properties of non-foamed PLA composites and discuss possible flame-retardant mechanisms. BC was used in place of natural bamboo powder, straw, rice husk or wood powder as a way of enhancing the stability and flame retardancy of the PLA. No interface compatibilizers or coupling agents were added to any of the BC/PLA composite materials, expecting that they would not be necessary and it would potentially help save on production costs. Part I covers the manufacture, physical characterization of constituent materials (particle size and surface morphology) and combustion tests—cone calorimeter test, UL-94 vertical flame test, and SEM images of combusted materials. Part II of this series covers the more in-depth elemental and thermal analyses before and after combustion of constituents and composites.

## 2. Materials and Methods

### 2.1. Materials

PLA was supplied by Cargill Dow Inc (4032D, Nature Works Co., Ltd., Blair, Nebraska, USA) with the following key properties: density: 1.24 g/cm^3^; melting point: 160 °C; glass transition temperature: 57.8 °C; and crystallization peak temperature: 150–170 °C. Bamboo powder (BP) and BC were supplied by Zhejiang Anji HuaShen Bamboo Charcoal Products Co., Ltd. (Zhejiang, China). BP was from milled moso bamboo and BC was obtained by pyrolizing fresh moso bamboo in a brick kiln at a carbonization temperature of 600 to 800 °C followed by grinding to a particle size of 50 to 200 mesh. AHP was supplied by Jinan Tai Xing New Material Co., Ltd. (Shandong, China); particle size ranged from 8 to 15 μm.

### 2.2. Preparation of PLA Composites

The blends were prepared using a double roll laboratory-scale compounder at 180 °C and 50 rpm for 10 min. Mix proportions are shown in Table 1.

Some blends were injection-moulded at 50 °C for 15 min into 80 × 10 × 4 mm^3^ sheets for cutting strips used in the UL94 and LOI tests, while the temperature of the cartridge was 180 °C. Other blends were hot-pressed into 100 × 100 × 3 mm^3^ sheets under 10 MPa and 180 °C for 10 min. for the cone calorimeter test (CCT), TG, TG-FTIR and XRD analyses. A schematic flow diagram of the manufacturing and testing procedures is shown in Figure 1. Adding BC above 25% adversely affected the mixing and the dispersion and uniformity of the material. Therefore, the addition of the AHP component was based on the mix BC/PLA (25/75).

### 2.3. Materials Characterization

*Particle size analysis* of BP and BC fillers was made using a laser particle size distribution instrument (HYL-1076, Dandong Haoyu Technology Co., Ltd., Shenyang, China) under liquid nitrogen conditions. The specific surface area and pore structure analysis of BC were carried out using the following steps using a specific surface area and porosity analyzer (ASAP2020, Micromeritics Co., Ltd., Norcross, Georgia, USA): (1) Setting the absorber pressure comparison number to 10 and keeping the weight of each sample about 0.35 g. (2) Degassing the samples for 3 h under vacuum at 120 °C before testing. (3) Setting target temperature to 200 °C for 180 min, and then cooling to room temperature to calculate actual mass. The sample tube containing the bamboo charcoal sample after cooling was transferred to the analysis station for the specific surface area test. The specific surface area and mesopore volume were calculated using the Brunauer–Emmett–Teller (BET) method, and the pore size distribution was calculated using the Density Functional Theory (DFT) method.

Differential Scanning Calorimetry (DSC) of the pure PLA was undertaken using a DSC-500B (Shanghai YANJIN Scientific Instrument Co., Ltd., Shanghai, China) calibrated with indium, tin and zinc standards. A sample of approx. 10 mg was weighed and hermetically sealed in an aluminum pan on a weighing scale with an empty pan used as the reference. The samples were heated from room temperature to 205 °C at a rate of 10 °C/min, held for 5 min, then cooled to 0 °C at a rate of 10 °C/min, then heated again to 180 °C at a rate of 10 °C/min.

*Elemental analyses* were carried out on the fabricated composites before cone calorimeter test using an Elemental Analyzer (Vario EL III, Elementar Co., Ltd., Langenselbold, Germany) in an oxidation furnace at 1200 °C for 10 min. Additionally, 0.2 ± 0.05 mg of BC was dried and placed in tin foil as a control.

*Mechanical properties* of the composites were tested by a microcomputer-controlled electronic universal testing machine (CMT610, MTS Industrial System (China) Co., Ltd., Shanghai, China). The tests followed the Chinese National Testing Standard GB/T 1040-2006 (for tensile properties) and GB/T 9341-2008 (for bending properties). The tensile test specimens were dumbbell-shaped with length, width and thickness being 60, 10 and 4 mm, respectively. The bending test specimens had length, width and thickness dimensions of 80, 10 and 4 mm, respectively. The gauge length of the tensile test was 60 mm and the loading speed 20 mm/min. The span for the bending test was 60 mm and the loading speed 5 mm/min. The flexural strength (MOR), stiffness (MOE) and Strain at Break (%) were averaged for six specimens tested per composite mix.

### 2.4. Cone Calorimetry and Limiting Oxygen Index (LOI) Tests

*Cone calorimeter combustion tests* were carried out using an ISO 5660 cone calorimeter (FTT 0007, Fire Testing Technology Co., Ltd., East Grinstead, UK). Test sample dimensions were 100 × 100 × 3 mm^3^. The appropriate test height was revised after wrapping the sample in aluminum foil and placing in the combustion chamber. The samples were exposed horizontally to 35 kW/m^2^ external heat flux. The combustion performance indices such as heat release and flue gas were recorded every 5 s until the flame was self-extinguished.

*Limiting oxygen index* (LOI) was tested according to ISO 4589-2:2017 and performed on a JF-3 oxygen index meter supplied by Jiangning Analysis Instrument Company (Jiangsu, China). The dimension of the test samples was 150 × 10 × 10 mm^3^. UL-94 vertical burning tests were performed using 5402-type vertical burning instrument (Suzhou Yangyi Woerqi Detection Technology Co., Ltd., Jiangsu, China). Specimen dimensions were 130 × 13 × 3 mm^3^, suspended vertically above a cotton patch. The classifications are defined according to the American National Standard UL-94 Vertical Flame Test.

The surface morphology of the constituents (BC, BP and AHP) and the carbonized residue morphology after cone calorimeter test were imaged using scanning electron microscopy (SEM; XL30E, Philips, Netherlands) at an accelerating voltage of 15 kV. Specimens were sputter-coated with 24 ct gold for 100 s at a current of 10 mA.

## 3. Results and Discussion

### 3.1. Characterization of BC, BP, AHP and PLA Materials

The elemental compositions of the BC (50–200 mesh) and BP are shown in Table 2. The chemical formula of BC fiber can be approximately expressed as (C_6_H_10_O_5_) _n_, with the main chemical components remaining to be lignin, cellulose and hemicellulose. The main element components of BC are carbon (C), hydrogen (H), nitrogen (N), sulfur (S), oxygen (O) and a relatively small amount of mineral ash (9.5%). The main component of BC is fixed carbon (>75%), its content being closely related to the raw material composition, carbonization time, carbonization temperature and water content of BC. C content of BC in other studies is around 80% [29]; the higher fixed carbon content as well as lower volatile component make BC an ideal material for polymer reinforcement [25,30]. In contrast, BP is comprised of mostly C (47.2%) and O (45.1%) and also contains higher amounts of H and O elements making it flammable under dry conditions, similar to wood flour or wood fiber used in WPC.

The estimated particle size distributions of BC and BP are shown in Table 3. The particle size distribution range for BC was mostly between 6 and 156 μm. Regarding BC powder, 10% was with particle diameters ≤ 5.6 μm, 50% with ≤ 22.7 μm, and 90% with ≤ 156.7 μm.

Figure 2a shows the pore size distribution of BC. Note that these pores refer to the nanometre-sized interstitial spaces found within the pyrolyzed cell walls of the bamboo tissue, which give BC its excellent water purification and gas molecule adsorption properties [31,32]. Studies by Asada et al. (2002), Wang et al. (2008) and Seo et al. (2016) [31,32,33] demonstrate how the specific surface area of bamboo charcoal can be increased with a reduced range of nano-pore sizes by adjusting the pyrolysis temperature between 400 and 1000 °C and in the atmosphere. These studies indicate optimum pyrolysis temperature and conditions varies according to the chemical compounds targeted for adsorption by the BC. The average nano-pore size here was 2.1 nm with most nano-pores falling within the 1–5 nm range. The nano-pores can be grouped into three categories (< 2 nm, which makes up most of the incremental pore volume in Figure 2a; 2–50 nm and > 50 nm, i.e., 0.05 µm). These pores do not include the macropores that make up the cell lumen and conductive vessel system of bamboo culm tissue, which are mostly in the 10–50 µm size range [30]. The estimated BET-specific surface area of the BC used here was 123.32 m^2^/g. Other studies [17] have estimated inner surface area of BC due to nano-porosity to be even higher; 250–390 m^2^/g, compared with 10 m^2^/g measured for wood charcoal. The very high nano-porosity of BC is believed to be able to absorb dissolved monomers or polymers in solution, such as dissolved LDPE or PVA, leading to an interlocked two-phase composite [25,34], although no direct evidence has been shown to support this. It may also greatly increase the external contact area of the BC particles with the PLA matrix improving the interfacial properties [35]. While unlikely, without further research it cannot be confirmed whether the melt blending process for PLA resulted in any PLA polymer infusing the nano-pore structure of the BC particles.

From Figure 2b, the glass transition temperature (T_g_) of PLA is 55 °C and cold crystallization temperature (T_cc_) was 118 °C, exhibiting a single crystalline melting peak of 148 °C. Figure 2c shows the characteristic absorptions of BC at 3509 cm^−1^, 1591 cm^−1^ and 1088 cm^−1^ corresponding to –OH, C=C, C–O bonds, respectively. In the FTIR curve of AHP, the peaks at the region of 1000–1100 cm^−1^ and 1100–1250 cm^−1^ are ascribed to the absorption vibration of P–O and P=O, respectively. The characteristic absorptions of BP at 2920 cm^−1^, 1431 cm^−1^ and 1029 cm^−1^ correspond to –CH, C–H and C–O, respectively. From Figure 2d, the characteristic absorptions of PLA at 2943 cm^−1^, 1758 cm^−1^, 1456 cm^−1^, 1088 cm^−1^ are assigned to alkyl and carbonyl groups. In the case of BP/PLA, BC/PLA, BC/PLA/AHP, main peaks observed are 3136 cm^−1^ (υ_–CH2–_), 2940 cm^−1^ (υ_–CH3–_), 1396 cm^−1^(υ_–O–C=O–_) and 1088 cm^−1^ (υ_C–O–C_) indicating that the BC was successfully introduced into the PLA matrix. It demonstrates that the BC consisted of both graphite-like carbon and amorphous carbon as shown schematically in Figure 2e. The surface and interior of BC have a large number of uniform fine hairs and pores, which are non-polar.

The surface morphologies of BC, BP and AHP particles are shown in Figure 3. There are obvious variations in the size and distribution, and shape of BC particles after mechanical crushing, consistent with the results from the particle size test. The ribbed carbonized fibers present in bamboo charcoal also have a rough surface which is beneficial for bonding with the PLA matrix [36]. AHP exhibits a comparably rough surface as well (Figure 3c), with further higher magnification showing that AHP is made up of clusters of multiple particles, which makes it easier to disperse during the melting process and has good compatibility with the PLA matrix.

### 3.2. Mechanical Properties

Figure 4 shows flexural properties of the composites at different BC mix ratios without AHP addition. The modulus of rupture (MOR), modulus of elasticity (MOE) and Strain at Break of pure PLA were 93.3 MPa, 2.9 GPa and 5.6%, respectively. Scaffaro et al. (2018) [37] and Ferreira et al. (2019) [38] showed that a part of external stresses can be absorbed by the fillers, while some are dissipated by particle-particle and particle-polymer interactions. They also studied a correlation between amount of filler, dispersion and improved mechanical properties of composites [39,40]. Similar to the previous findings, the flexural strength of the PLA decreased with increasing dosage of BC filler, but at the same time, the material became stiffer with *E* increasing. After about 30% addition, the decline in mechanical strength tapered off. The observed loss in flexural strength was likely due to poor interfacial adhesion and hence low stress transfer between PLA matrix and BC [41]. A cross-linking agent may be required to rectify this. Excessive BC content >25 wt. % caused problems with agglomeration in the PLA matrix, forming weak interfaces and affecting stress transfer. This reduction is probably related to a weak interaction between the BC and PLA. Along with flexural strength, the flexural strain at break was also reduced after the addition of BC. Figure 5 shows the effect of different levels of AHP addition to an optimal base mix of 25% BC and 75% PLA. The baseline MOR, MOE and Strain at Break (%) were 45.3 MPa, 4.1 GPa and 1.3%, respectively. Increasing the level of AHP reduced the PLA in the composite from 75 down to 45%, but despite the reduced ratio of polymer and increased ratio of filler, the addition of AHP further improved the mechanical properties of the PLA/BC mix.

From Figure 5, the MOE was significantly improved, particularly at 25 wt. % AHP, increasing to 5.3 GPa (29% higher than that without AHP). The MOR with 25% AHP was 68.7 MPa, 34% higher than that of BC/PLA (25/75). However, MOR declined as the addition of AHP exceeded 25 wt. %, likely as the combined increased volume of fillers excessively reduced the quantity and continuity of polymer matrix. Non-uniform pores and inclusions can be the defects leading to fracture. With the formation of sub-cracks [17], the bending strain at the fracture decreases with the decrease in bending strength. When AHP was added at 15%, the elongation at break was the best, and the further addition of AHP increased the risk of non-uniformity in the preparation of composite materials, resulting in a significant decrease in the ductility of composites. AHP likely interacted with the surrounding BC to produce a larger specific surface area, thus promoting stress transfer from the BC/AHP mixture to PLA [25]. AHP is believed to increase the interfacial adhesion between BC and PLA, improving the flexural properties of the composites, which is beneficial for the development of flame-retardant PLA-based composites.

### 3.3. Early Burning Behavior and Flame Retardancy (LOI and UL94 Vertical Flame Tests)

LOI and vertical burning tests are widely used to evaluate the flammability of materials [42]. Results are shown in Table 4. Pure PLA normally has a low LOI value of around 19 vol% [43]. The pure PLA measured here had a LOI value of 22.2 vol%, and the flame retardancy of BC/PLA (25/75) composites was only slightly improved (LOI value: 23.8 vol%), which may be due to the formation of char on the surface of the composite. The LOI value of the BC/PLA composite increased steadily with increasing AHP addition, to 31 vol% with 30 wt. % AHP addition. LOI observations are also influenced by the declining content of flammable PLA in the test material as AHP content increases.

Pure PLA exhibited significant melt dripping during the UL-94 vertical burning test. With 10% AHP, the LOI increased to 26 vol%, passable at V-0 level under the UL-94 test, and with 20% AHP, the LOI reached 29.3 vol%, 23% higher than that of the BC/PLA (25/75) mix, and 32% higher than pure PLA. The composite containing AHP could be rated as a secondary non-combustible material. In the vertical burning test, the BC/PLA mix containing 10% or more AHP showed no melt dripping of the PLA polymer. The addition of AHP appears to inhibit the melting of the polymer and significantly improves the flame-retardant properties of the BC/PLA composites. Images of the materials after the vertical burning test are shown in Figure 6.

To evaluate the interaction between BC and AHP [44], the synergistic effect *Es* is introduced as follows:*Es*={(*F_p_*)_[fr + s]_−(*F_p_*)_p_}/{(*F_p_*)_fr_−(*F_p_*)_p_) + ((*F_p_*)_s_−(*F_p_*)_p_}(1)
where (*F_p_*)_p_ is the flame-retardant property of the polymer alone, (*F_p_*)_fr_ refers to the polymer containing flame retardant, (*F_p_*)_s_ is defined as a polymer with a synergist, and (*F_p_*)_[fr + s]_ is that polymer with a synergy system. The synergistic flame-retardant effect between BC and AHP occurs if *Es* > 1. Since the LOI values are closely related to both UL94 ratings and cone calorimeter parameters [45], LOI values are assumed for *Fp* in Equation (1). The results are tabulated in Table 5. BC and AHP have a good synergistic effect on the flame retardancy of PLA, as evidenced by the corresponding *Es* being greater than 1 and increasing with the addition of AHP.

Figure 6a shows the melt dripping behaviour of the BC/PLA (25/75) mix. With the addition of AHP (Figure 6b–f) melt dripping almost completely disappeared and a carbonized layer was formed at the top of the sample to delay or eliminate the upward spread of the burning flame. AHP significantly increases the melt viscosity of the composite [25]. Previous work by Liu et al. (2016) [41] found that adding AHP on its own up to 20% cannot completely suppress melt dripping of PLA. However, Tang et al. (2012) [26] was able to completely suppress melt dripping during the UL 94 test by adding 20% AHP to PLA. Nevertheless, the BC powder plus the AHP appears to have a strong synergistic effect to more effectively suppress the melt dripping and flame spread of PLA, improving its flame-retardant properties. The addition of 5% AHP is insufficient with the LOI value of the mix being similar to pure PLA at 23.4 vol%, reaching only UL 94 V-2 rating. At 20% AHP addition to the BC/PLA (25/75) mix, the LOI exceeded 29 vol%, giving it a UL 94 V-0 rating.

### 3.4. Cone Calorimetery Testing

The cone calorimeter is based on the oxygen consumption principle, more accurately simulating the combustion of materials in a real fire situation, and is commonly used for testing new flame-retardant materials [46,47,48]. The appearances of cone calorimeter specimens before and after cone calorimeter test are shown in Figure 7. The corresponding data are given in Table 6 and the key combustion indicators with time graphs are shown in Figure 8.

From Figure 7, the BP/PLA composite underwent almost complete combustion, leaving little residue. The addition of BC and particularly BC/AHP mix allowed for good material retention after the cone calorimeter via the formation of a protective layer of char. Figure 8c_1_ to c_4_ provide a good visual summary of the relative effects on combustion of adding just BP or BC, or adding AHP to the system. For PLA containing BC, BP, BC+AHP, the ignition time (TTI) was shorter than pure PLA, indicating that the thermal stability of the composite is reduced.

Heat release rate (HRR) is the rate at which heat is released per unit area of specimen surface. The greater the value, e.g., for pure PLA, the more heat is released back to the surface of the material, accelerating the pyrolysis and flame propagation rates of the material and generation of volatile combustibles. From Table 6 and Figure 8a_1_, the addition of 25% BC decreased the peak heat release rate (pHRR) of the composite to 320.32 kW/m^2^ compared to pure PLA (484.27 kW/m^2^). At 40% BC, pHRR was reduced to 267.07 kW/m^2^ or 55.15% of pure PLA. From Figure 8c_1_ adding BP reduced HRR slightly but was not as effective as BC or the much more effective BC/AHP combination. pHRR was lowest for the BC/PLA/AHP (25/50/25) system at 130.77 kW/m^2^, see Figure 8b_1_,c_1_. The HRR-time curves for the composites are characteristic of “thick charring (residue forming) materials” defined by Shartel and Hull (2007) [47] as undergoing an initial increase in HRR until an efficient char layer is formed. As the char layer thickens, this results in a decrease in HRR. The maximum reached at the beginning equals both the average or steady HRR, and the pHRR. For flame-retarded PLA nanocomposites there is a positive relation between reduction in pHRR during cone calorimeter test and the char yield [24]. Char residue acts as a thermal barrier for both external heat flux and heat feedback, thus effectively protecting the underlying material and improving the flame retardancy of the PLA composite.

Adding BP or BC adjusted the total heat release (THR)-time curve (Figure 8a_2_) somewhat; 40% BC addition reduced THR to 53.89 MJ/m^2^. The effect of the co-addition of AHP can be clearly seen from Figure 8b_2,_c_2_. Co-addition of 30% AHP further reduced THR 45.11 MJ/m^2^. Note THR of BP/PLA (25/75) was slightly higher (78.01 MJ/m^2^) than pure PLA or the BC+AHP mixes. Surprisingly, the THR of the composites increased slightly when 20 and 25% AHP was added. This may be explained by the fact that THR is calculated based on oxygen consumption, and the thermal degradation of AHP consumes oxygen, resulting in a small increase in the registered THR value.

Effective heat of combustion (EHC) refers to the heat released by the combustion of volatiles formed by thermal decomposition of the material. It is used to measure the flammability and degree of combustion of pyrolysis gasification products. The lower the EHC value, the better the flame-retardant properties of the material when exposed to high heat. EHC over the first 100–200 s is relatively high for the BC- and BP-added PLA but then drops below pure PLA as combustion continues (Figure 8a_3_), particularly for BC. The BC composites are more porous than pure PLA and trace flammable gases are produced early at the surface as BC stimulates the thermal degradation mechanism of PLA. AHP significantly reduced the pEHC over the first 100–200 s of combustion compared to adding only 25% BC (Figure 8c_3_). 

Specific extinction area (SEA) refers to the amount of smoke produced per unit mass of specimen. From Table 6 the addition of BC and BP significantly increased the pSEA of the composites from 79.5 m^2^/kg for pure PLA to 1160 and 1538 m^2^/kg in some cases, e.g., BC/PLA (40/60) and BC/PLA (25/75). This finding would be of significant concern for smoke generation and toxicity to occupants and fire-fighters during a fire involving a large volume of these materials. Adding AHP decreased the pSEA of the composites compared with the addition of BC only, but the pSEAs of these mixes were still higher than pure PLA. Note that increasing the AHP content also increased pSEA to close to 1000 m^2^/kg.

As a general rule, polymers with an aliphatic backbone, like PLA, tend to generate less smoke during combustion than polyenic polymers and those with pendant aromatic groups, such as polystyrene, which produce more smoke [49]. Organic materials produce toxic carbon monoxide (CO) when not completely combusted, and in the later stages of combustion CO is oxidized to CO_2_ [50]. From Table 6, the addition of BP significantly increased the amounts of CO and CO_2_ released from the material, whereas its substitution with BC had very mixed results for CO and CO_2_ release. For example, BC/PLA (25/75) had CO and CO_2_ yields of 0.01 and 105.59 kg/kg, whereas BC/PLA (30/70) had CO and CO_2_ yields of 2.77 and 6327.12 kg/kg. The CO and CO_2_ yields and balances are discussed further in Part II. Surprisingly, the addition of AHP did not significantly affect CO and CO_2_ release.

Mass loss rate (MLR) and peak mass loss rate (pMLR) reflect the rate and extent of pyrolysis of the material under thermal radiation. From Figure 8a_4,_c_4_, the addition of BC increased pMLR of the composite up to 23.4% for BP/PLA, but the addition of AHP significantly reduced pMLR of the BC/PLA composite. In Figure 8b_4_, pMLR was 0.29 g/s, a reduction of 38% from pure PLA and 49% from BC/PLA (25/75). The residue mass (RM%) represents the proportion of the combustion residue (residual carbon) left over from the original mass. From Figure 8a_5,_ c_5_, the RM increased with increasing BC addition, following the trend of pMLR. The BP/PLA mix was lower in RM, reflecting its greater content of combustible organic matter. Adding AHP to the BC/PLA mix increased the RM values with increasing levels of AHP addition as shown in Figure 8b_5_. At 30% AHP the RM was 2.39 times that of the BC/PLA (25/75) composite.

For further assessment of pyrolysis behavior, the Fire Performance Index (FPI) and the Fire Growth Index (FGI) were used to compare the flame-retardant properties of the composites. FPI is the ratio of TTI and the pHRR, and there is a correlation between the FPI of a material and the time to flashover [51,52]. The greater the FPI value, the greater the window for safe evacuation. FGI is the ratio of pHRR and the time to pHRR—the higher the FGI value, the greater the risk of the material causing rapid conflagration in the event of a fire. From Table 6, FPI of the various BC/PLA mixes changed little from 0.10. The FGI of BP/PLA was highest at 3.05, but adding BC also increased FGI compared to pure PLA. BC addition promotes early thermal degradation, decreasing the time to pHRR. Note that FPI increased to 0.20 with 30% AHP addition—nearly twice that of pure PLA. Correspondingly, FGI was unchanged or lower (2.07 for 25% AHP), reducing the flame spread risk in the event of a fire. The results demonstrate the effectiveness of AHP as a flame retardant by effectively reducing pHRR through the early formation of a protective char layer.

It should be noted that the results from cone calorimeter tests and early burning behaviour (LOI and UL-94) are often not correlated with one another, but rather provide complementary information on different stages of the combustion process [53,54]. Cone calorimeter results for intumescent flame-retarded PLA from Wei et al. (2011) [40] found no significant differences in char yields, heat release rate, and total heat release rate, etc., for neat PLA and flame retarded PLA containing a high P-content oligomer WLA-3 flame retardant. However, there were significant differences observed in their early burning behaviours from the LOI and UL-94 melt and flame test.

### 3.5. SEM Images of Residues 

SEM was used to examine the raw (Figure 9) and carbonized (charred) structures formed during the cone calorimeter test (Figure 10). Figure 9a displays a fractured surface in a low-magnification image. In Figure 9b, a loose piece of BC can be seen in the upper LR corner. The smooth mass in the middle is PLA with some air pockets likely introduced during melt blending. A granule of AHP is circled, along with what is believed to be an embedded piece of BC within the PLA matrix.

In Figure 10a_1_–c_1_, the outer char layer of the BC/PLA/AHP (25/45/30) composite was much more compact and continuous than in the other mixes containing less AHP. Similarly, SEM images from Tang et al. (2012) and Gu et al. (2019) [26,55] observed a compact structure of the char layer formed on PLA-AHP composites with bubbles suggested to be formed in the PLA by gasses released from decomposition of AHP during combustion. The compact char layer plays a very important role in protecting the polymer from further heat and oxygen transfer in, escape of flammable gasses, melting and flame spread [26,56]. During thermal degradation of the mixes with lower AHP levels, a more friable crust was formed in the outer matrix which was believed to reduce its flame-retardant action. Interestingly, the particle size of carbonized materials seemed to increase at higher levels of AHP addition. Observation of internal layers of combustion residues (Figure 10a_4_–c_4_) shows that the BC was integrated well and uniformly with the AHP. The surface and surface pores of the carbon residue were observed by SEM, but the inner layers of the test sample were not imaged. Note in Figure 10c_4_ a piece of BC that has retained the structure of a bamboo parenchyma cell wall with its original small oval pits.

Adding AHP significantly reduced total mass loss. The AHP flame-retardant system forms a low-stability compound by promoting the early, partial thermal degradation of PLA to form a surface layer of char, which significantly reduces further thermal degradation of inner material, improving the flame retardancy of the composite. It seems that BC/AHP mix also promotes early surface degradation, forming solid-phase products that do not cause weight loss and are able to block heat and oxygen. The likely mechanisms of the BC and AHP flame retardant systems are discussed in further detail in Part II.

## 4. Conclusions

A set of BC/PLA/AHP composites were produced using standard WPC compounding and melt blending processes. BC addition on its own reduced the MOR and Strain at Break while the PLA composite became more brittle with higher MOE and earlier fracturing. Adding AHP to a base mix of 75% PLA and 25% BC appeared to improve mechanical properties, possibly by enhancing the interfacial adhesion between BC and PLA resulting in improved flexural properties of composites, despite lower PLA content. The LOI of the BC/PLA/AHP (25/55/20) composite exceeded 29 vol%, passing the UL-94 V-0 and effectively eliminating melt dripping of the PLA polymer. During the UL-94 Vertical Flame Test, the surface material of the PLA with BC+AHP degraded early to form a carbonized protective crust typical of an intumescent flame-retardant system. From the cone calorimeter results adding only BC to PLA accelerated the HRR, THR, EHC and MLR but reduced their total extent over the course of the cone calorimeter test. The addition of BP and BC significantly increased the specific extinction area (SEA) of PLA, but the addition of AHP helped substantially mitigate the SEA. In addition, the combination of BC and AHP was more beneficial to reducing the generation of CO and CO_2_ than that of BP and AHP. Further addition of 15–30% AHP drastically reduced all combustion indices, and increased the RM after cone calorimeter test. 

Thermal degradation of the surface BC/PLA/AHP composites started earlier with further addition of AHP, which at 30% addition significantly increased the protective carbon residue or ‘char’ production rate to 40.3%, or 43 times that of pure PLA. In accord with previous studies on adding AHP to PLA, SEM images showed larger and more continuous carbonized structures, believed to effectively impede the transfer of heat, oxygen and pyrolysis gases through the protective char layer. 

The BC addition levels, up to 40%, were considerably higher than 8% reported in previous works, which helped boost the flame retardancy of PLA composites. However, the effects of such high levels of these non-reinforcing fillers on composite functional properties for its end use need to be further verified.

## Figures and Tables

**Figure 1 polymers-12-02217-f001:**
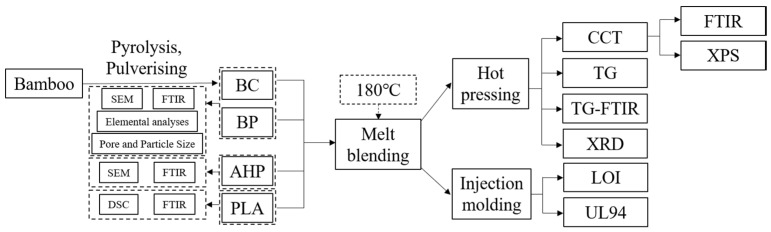
Flowchart diagram showing preparation and tests of the composites.

**Figure 2 polymers-12-02217-f002:**
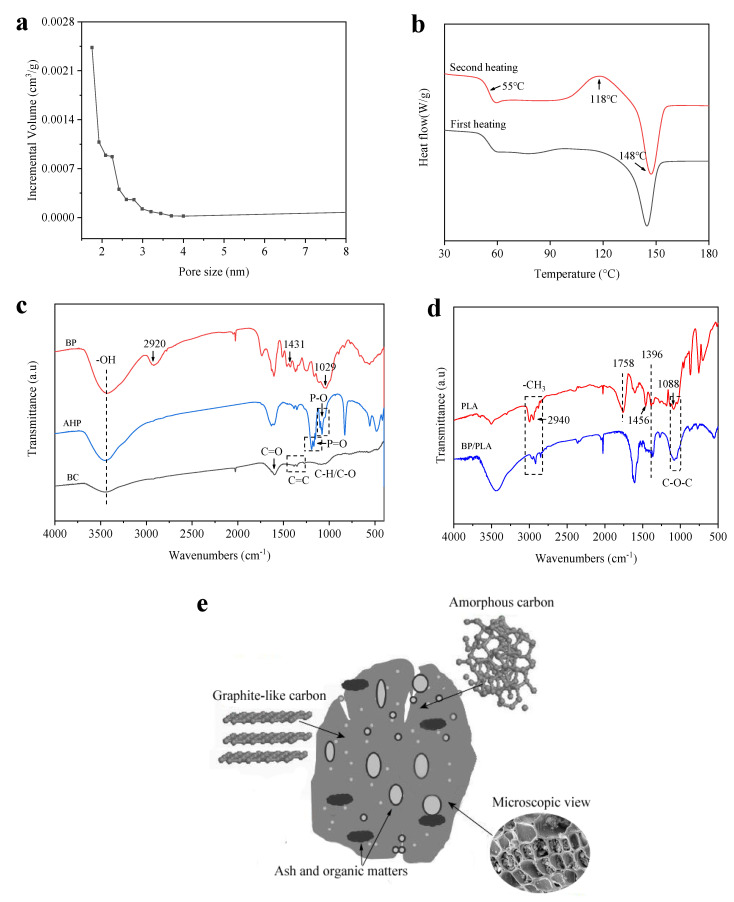
(**a**) Pore size distribution of BC, (**b**) DSC curves for PLA, (**c**) and (**d**) FTIR spectra of BC, BP, AHP, PLA, BP/PLA (25/75), BC/PLA (25/75) and BC/PLA/AHP (25/50/25) and (**e**) schematic diagram of BC pore structure.

**Figure 3 polymers-12-02217-f003:**
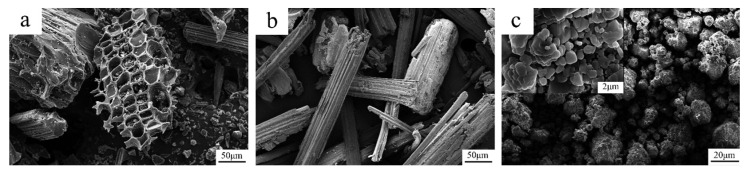
SEM images of (**a**) BC, (**b**) BP and (**c**) AHP.

**Figure 4 polymers-12-02217-f004:**
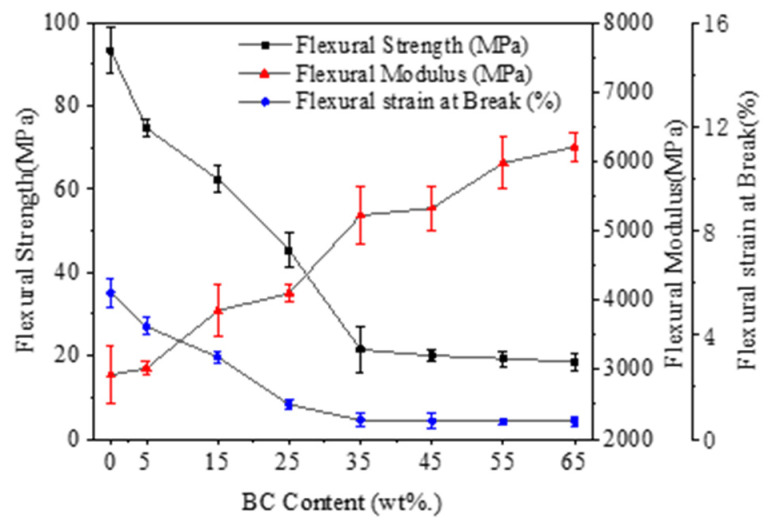
Flexural property curves of BC/PLA composites.

**Figure 5 polymers-12-02217-f005:**
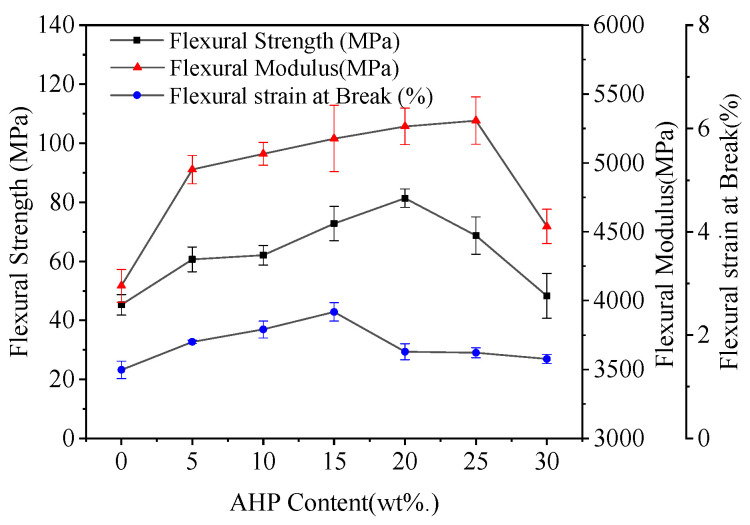
Flexural property curves of BC/PLA/AHP.

**Figure 6 polymers-12-02217-f006:**
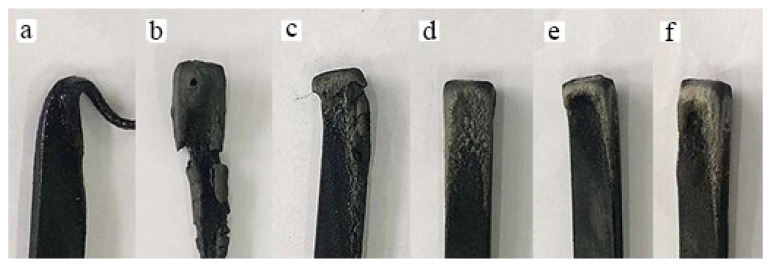
Images of BC/PLA composites after UL94 test: (**a**) BC/PLA (25/75), (**b**) AHP/BC/PLA (25/70/05), (**c**) AHP/BC/PLA (25/65/10), (**d**) AHP/BC/PLA (25/60/15), (**e**) AHP/BC/PLA (25/55/20) and (**f**) AHP/BC/PLA (25/50/25).

**Figure 7 polymers-12-02217-f007:**
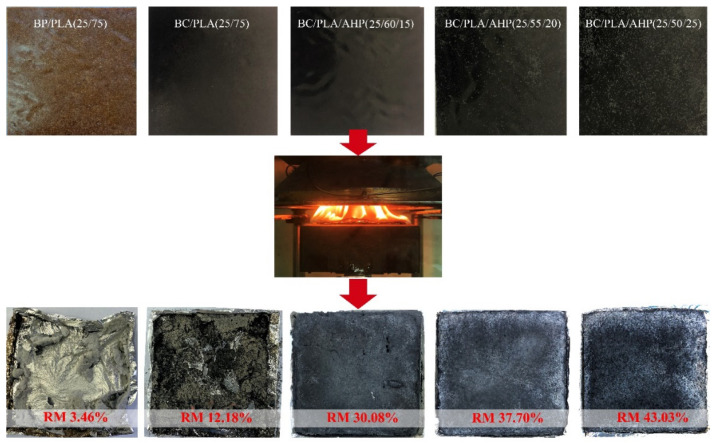
Appearance of test materials before and after cone calorimeter test.

**Figure 8 polymers-12-02217-f008:**
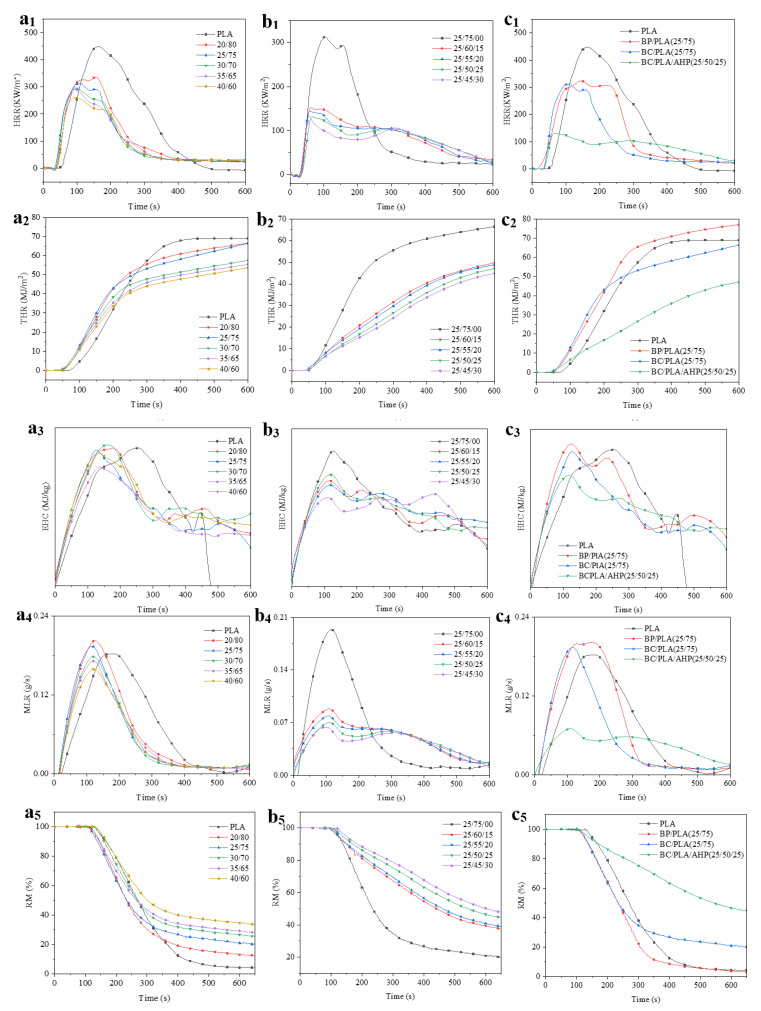
Heat release rate (HRR), total heat release (THR), effective heat of combustion (EHC), mass loss rate (MLR) and residue mass (RM) curves for: (**a_1_–a_5_**) BC/PLA, (**b_1_–b_5_**) BC/PLA/AHP and (**c_1_–c_5_**) BP/PLA.

**Figure 9 polymers-12-02217-f009:**
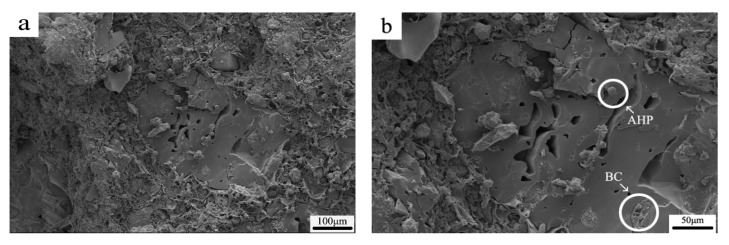
SEM images of fractured surface of BC/PLA/AHP (25/50/25) composites before combustion: (**a**) 100 μm; (**b**) 50 μm.

**Figure 10 polymers-12-02217-f010:**
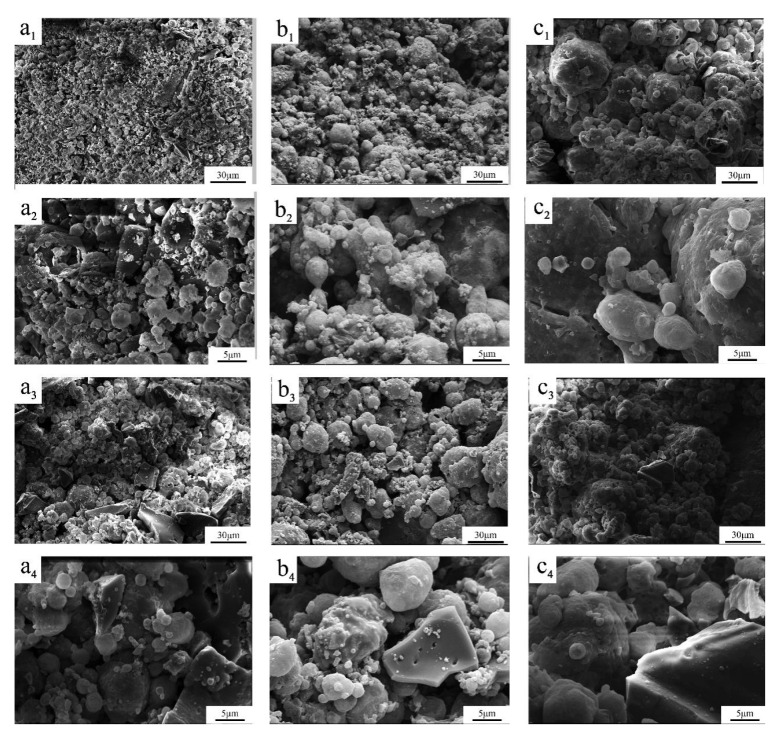
SEM images of char residue on surface and surface pores after cone calorimeter test with different magnification: (**a_1_–a_4_**) BC/PLA/AHP (25/55/20), (**b_1_–b_4_**) BC/PLA/AHP (25/50/25), and (**c_1_–c_4_**) BC/PLA/AHP (25/45/30).

**Table 1 polymers-12-02217-t001:** The formulation of BP/PLA), (BC)/PLA, BC/PLA/AHP composites.

Sample	Composition (wt. %)			
	PLA	BP	BC	AHP
PLA	100	0	0	0
BP/PLA (25/75)	75	25	0	0
20BC/PLA (20/80)	80	0	20	0
25BC/PLA (25/75)	75	0	25	0
30BC/PLA (30/70)	70	0	30	0
35BC/PLA (35/65)	65	0	35	0
40BC/PLA (40/60)	60	0	40	0
BC/PLA/AHP (25/70/5)	70	0	25	5
BC/PLA/AHP (25/65/10)	65	0	25	10
BC/PLA/AHP (25/60/15)	60	0	25	15
BC/PLA/AHP (25/55/20)	55	0	25	20
BC/PLA/AHP (25/50/25)	50	0	25	25
BC/PLA/AHP (25/45/30)	45	0	25	30

**Table 2 polymers-12-02217-t002:** Chemical composition of BC and BP.

Content (wt. %)	C	H	N	S	O	Fixed Carbon	Volatile Component	Ash
BC	75.7	2.9	0.7	0.2	11.0	75.7	14.8	9.5
BP	47.2	6.2	0.1	0.3	45.1	16.5	82.5	1.1

**Table 3 polymers-12-02217-t003:** Particle size distribution of BC and BP.

Material	Dx (10)/μm	Dx (50)/μm	Dx (90)/μm
BC	5.6	22.7	156.7
BP	7.	54.8	192.5

**Table 4 polymers-12-02217-t004:** Effect of AHP on the flame retardancy for BC/PLA/AHP composites.

Sample	w(AHP)/%	LOI/%	UL-94		
			t_1_/t_2_	Dripping	Rating
1	0	23.8	3.3/2.3	Yes	V-2
2	5	24.0	5.1/1.4	Yes	V-2
3	10	26.0	3.1/1.6	No	V-0
4	15	27.1	1.0/1.2	No	V-0
5	20	29.3	1.0/1.1	No	V-0
6	25	30.0	1.0/1.6	No	V-0
7	30	31.0	1.2/1.1	No	V-0

**Table 5 polymers-12-02217-t005:** Synergistic effect of BC and AHP on PLA.

Sample	LOI/vol%	Syn Effectivity
PLA	22.0	-
BC/PLA (25/75)	23.8	-
PLA/AHP (75/15)	25.2	-
PLA/AHP (75/20)	26.4	-
PLA/AHP (75/25)	26.8	-
PLA/AHP (75/30)	27.3	-
BC/PLA/AHP (25/60/15)	27.1	1.02
BC/PLA/AHP (25/55/20)	29.3	1.18
BC/PLA/AHP (25/50/25)	30.0	1.21
BC/PLA/AHP (25/45/30)	31.0	1.27

**Table 6 polymers-12-02217-t006:** Cone calorimeter data for PLA, BP/PLA, BC/PLA and BC/PLA/AHP composites.

Sample	TTI (s)	pHRR (kW/m^2^)	THR (MJ/m^2^)	pMLR (g/s)	RM (%)	pEHC (KJ/kg)	pSEA (m^2^/kg)	Peak COY/CO_2_Y kg/kg	FPI	FGI
PLA	51	484.27	68.97	0.47	4.37	73.41	79.53	0.19/1420.79	0.11	2.56
BP/PLA (25/75)	38	330.05	78.01	0.77	3.46	79.52	333.43	0.33/2181.69	0.12	3.05
BC/PLA (20/80)	38	342.47	66.78	0.56	12.18	78.54	747.55	0.10/242.21	0.11	2.25
BC/PLA (25/75)	33	320.32	59.94	0.57	19.46	79.84	1538.26	0.01/105.59	0.10	2.44
BC/PLA (30/70)	27	298.66	57.77	0.58	24.69	77.18	500.54	2.77/6327.12	0.09	2.99
BC/PLA (35/65)	27	296.09	55.94	0.54	26.80	77.06	740.01	0.03/177.54	0.09	3.12
BC/PLA (40/60)	27	267.07	53.89	0.43	32.38	79.17	1159.29	0.18/478.02	0.10	2.90
BC/PLA/AHP (25/60/15)	27	155.01	47.96	0.40	30.08	78.24	330.52	0.01/245.41	0.17	2.54
BC/PLA/AHP (25/55/20)	27	144.14	49.73	0.33	37.70	70.83	649.82	0.17/308.66	0.18	2.25
BC/PLA/AHP (25/50/25)	27	130.56	48.19	0.29	43.03	72.52	369.71	0.26/409.17	0.21	2.07
BC/PLA/AHP (25/45/30)	27	134.77	45.11	0.33	46.59	76.94	977.52	0.01/258.21	0.20	2.50

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
