# Peer review of "Development of Biodegradable Flame-Retardant Bamboo Charcoal Composites, Part I: Thermal and Elemental Analyses"

_polymers, 2020, doi:10.3390/polym12102217_

Round 1

Reviewer 1 Report

Dear Authors,

the study concerning the development and the properties of flame-retardant bamboo charcoal composites is interesting, well organized and well documented.

The introduction presents a short overview of the different points that will be addressed in the publication. Nevertheless, two points are introduced in the title or in the introduction  (I mean: biodegradability and environmental concerns) without further discussion in the publication. Have you made some tests on your composites to discuss these 2 points?

Concerning the results:

  1. You give the chemical composition of BC and BP. It is interesting but can you add the typical formula of such particles (like for the formula that we can find for lignin, etc.)
  2. In page 8, figure 2.d, you only show FTIR curve of BP/PLA composite to explain a good introduction of such additive in PLA. Could you add the FTIR curves for the other composites, i.e. for those containing BC or BC/AHP? Furthermore, in page 9, line 271 you conclude that the FTIR results indicate a successful mixing. Complementaries microscopic analyses are needed to support this conclusion. For example, SEM analyses are necessary to confirm the agglomeration of particle for higher content than 25 wt.% as mentioned on page 9, line 288-289.
  3. You indicate an improvement of flexural properties with higher AHP content with BC in PLA. But, it really lowers than pure PLA. Could you add a comment about that and the impact for potential applications? Why did you choose the flexural properties rather than the tensile ones?
  4. In page 10, line 301, it is mentioned: "AHP is believed to increase the interfacial adhesion between BC and PLA, improving the flexural properties of the composites [39]". Could you give more details about the better adhesion; what are the mechanisms behind this conclusion? Maybe I am wrong, but I didn't read anything about this conclusion in reference 39 which is dedicated to PS and not to PLA!
  5. In page 11, you explain the UL-94 results by a modification of the viscosity. It is a possibility that you justify by a reference found in the literature. Have you made some viscosity measurements on your own measurements to confirm this phenomenon with BC?
  6. Always on page 11, but in line 342, you write that there is a strong synergy between AHP and BC. Nevertheless, in line 338, it is confirmed that there is no PLA/AHP composite for the control. So, how can you conclude on a synergy if the effect of each independent component has not been evaluated? 
  7. Concerning the comment made in line 456 on page 17, I do not totally agree. It is written:"...more compact and continuous than in the other mises containing less or no AHP." OK for the first part but there is no image justifying the comment for no AHP.
  8. The conclusion should repeat the remarks made on the smoke production because it is a big concern.
  9. Minor remarks:
  10. the definition of WPC on page 3 is given after its first use
  11. IFR is not explained
  12. typing errors: line 106 (CaCO3); line 185 (space between unit and value); line 282 (modulus of rupture and not rapture: or change by maximal flexural strength); check the labelling of figure 6 (15wt.% of PLA !!!); line 393 BP and not PB
  13. Figures 2e) and 9a) are not exploited
  14. Please add the standard deviations for all the results (mechanical and fire)

Author Response

Point 1: The introduction presents a short overview of the different points that will be addressed in the publication. Nevertheless, two points are introduced in the title or in the introduction (I mean: biodegradability and environmental concerns) without further discussion in the publication. Have you made some tests on your composites to discuss these 2 points?

Response 1: It is well known that PLA and BC are both biodegradable, compostable (under correct conditions) and environmentally friendly materials. The composites prepared by melting blending of the two without additives are also environmentally friendly and biodegradable.

Point 2: You give the chemical composition of BC and BP. It is interesting but can you add the typical formula of such particles (like for the formula that we can find for lignin, etc.)

Response 2: BC is mainly composed of bamboo charcoal fiber. Chemical formula can be approximately expressed as (C6H10O5)n. The chemical composition is mainly lignin, cellulose and hemicellulose.  The three belong to the high polysaccharide, the total amount of which accounts for more than 90% of the dry fiber, followed by protein, fat, pectin, tannin, pigment, ash.  Therefore, BC has no specific chemical formula. See Lines: 212-214.  

Point 3: In page 8, figure 2.d, you only show FTIR curve of BP/PLA composite to explain a good introduction of such additive in PLA. Could you add the FTIR curves for the other composites, i.e. for those containing BC or BC/AHP? Furthermore, in page 9, line 271 you conclude that the FTIR results indicate a successful mixing. Complementaries microscopic analyses are needed to support this conclusion. For example, SEM analyses are necessary to confirm the agglomeration of particle for higher content than 25 wt.% as mentioned on page 9, line 288-289.

Response 3: We have added the FTIR curves for BC/PLA and BC/PLA/AHP composites in page 8, figure 2.d. The infrared results obtained on page 9 can only indicate that BC is successfully introduced into the PLA matrix, but the result of mixing BC and PLA uniformly cannot be obtained. The infrared analysis part has been corrected, and AHP and BC are embedded in the PLA matrix as seen from the SEM diagram of BC/PLA/AHP(25/50/25) fracture. See Lines: 301-306.

Point 4: You indicate an improvement of flexural properties with higher AHP content with BC in PLA. But, it really lowers than pure PLA. Could you add a comment about that and the impact for potential applications? Why did you choose the flexural properties rather than the tensile ones?

Response 4: The bending strength of BC/PLA/AHP is higher than that of BC/PLA composite. The paper of Ferreira et al (2019) was added, supporting that due to the weak interaction between BC and PLA, the strength of the material was still lower than that of pure PLA. See Lines: 320-324.

Tensile strength is an intrinsic value of the material, independent of shape. Bending strength refers to the ability of the material to resist bending and not break. It is more used to investigate the strength of brittle materials. The bending strength is also influenced by more factors, such as thickness or resin content, to which tensile strength is less sensitive. It is well known that PLA is a brittle material, and the addition of filler will further deteriorate its mechanical properties under normal circumstances. The bending property was selected, mainly to consider the nucleation effect of BC. On this basis, the influence of AHP on PLA brittleness was explored.

Point 5: In page 10, line 301, it is mentioned: "AHP is believed to increase the interfacial adhesion between BC and PLA, improving the flexural properties of the composites [39]". Could you give more details about the better adhesion; what are the mechanisms behind this conclusion? Maybe I am wrong, but I didn't read anything about this conclusion in reference 39 which is dedicated to PS and not to PLA!

Response 5: This mechanism can be assumed that AHP collaborates with the surrounding BC to produce a larger specific surface area, thus promoting the stress transfer from the

BC/AHP mixture to PLA. We changed the reference to Ref 23 (Mousa et al 2018) and deleted Ref 39. See Lines: 344-348.

Point 6: In page 11, you explain the UL-94 results by a modification of the viscosity. It is a possibility that you justify by a reference found in the literature. Have you made some viscosity measurements on your own measurements to confirm this phenomenon with BC?

Response 6: Viscosity measurements are usually made using a rotary viscometer at room temperature for a liquid resin. When verifying the viscosity of the polymer, it should be measured in its molten state. At temperature higher than 170℃, we could not measure viscosity.  Hence only reference was used.

Point 7: Always on page 11, but in line 342, you write that there is a strong synergy between AHP and BC. Nevertheless, in line 338, it is confirmed that there is no PLA/AHP composite for the control. So, how can you conclude on a synergy if the effect of each independent component has not been evaluated?

Response 7: We added the reference Lewin (2001, Ref 51) and Horrocks (2010, Ref 52) paper, according to polymer separate flame retardant performance, containing flame retardant polymer flame retardant performance, containing synergist flame retardant properties of polymer and flame retardant properties of polymer has the association effect system, using the formula of BC and to quantify the synergistic effect of AHP, the results show that the higher the content of AHP, synergistic effect, the better.  See added Lines: 381-413.

Point 8: Concerning the comment made in line 456 on page 17, I do not totally agree. It is written:"...more compact and continuous than in the other mises containing less or no AHP." OK for the first part but there is no image justifying the comment for no AHP.

Response 8: We deleted the “or no” in the analysis in line 546 on page 18, highlighting the relationship between the morphology of carbon residue under high AHP content and its flame retardancy.

Point 9: The conclusion should repeat the remarks made on the smoke production because it is a big concern.

Response 9: The remark on smoke generation has been added to the conclusion. See Lines:  577-580.

Point 10: the definition of WPC on page 3 is given after its first use IFR is not explained

Response 10: The definition of WPC is removed from page 3 and explained by IFR. See Lines 119 and 121.

Point 11: typing errors: line 106 (CaCO3); line 185 (space between unit and value); line 282 (modulus of rupture and not rapture: or change by maximal flexural strength); check the labelling of figure 6 (15wt.% of PLA !!!); line 393 BP and not PB

Response 11: The above typing errors have been modified in the text. line 107 (CaCO3) to CaCO3; line 187 (space between unit and value) was deleted; line 319 (rapture) to rupture; the labelling of figure 6 (15wt.% of PLA) to AHP/BC/PLA (25/70/1505) in line 415; line 473 PB to BP.

Point 12: Figures 2e) and 9a) are not exploited

Response 12: We exploited Figures 2e) and 9a). See Lines: 306-307, 530.

Point 13: Please add the standard deviations for all the results (mechanical and fire)

Response 13: The mechanical and fire standard deviation was included. See Lines: 185-186, 193.

Reviewer 2 Report

The research article poresented by authors are very interesting, it has a new flame retardant combination and the system is very well studied, it will attract the young researchers, environmentalists, and polymer composite engineers.

I would like to bring some modifications and they are the authors should reduce the introduction to half, make it concise, in the line 59 unneceesarilly extra character please remove it. Do not use CCT acronym for cone calorimetry, it is known as cone calorimetry. In conclusion more emphasis should be on the success of the research and should indicate where these materials will find their use.

Finally the reviewer appreciates the detailed research of these innovative research.

Author Response

Point: I would like to bring some modifications and they are the authors should reduce the introduction to half, make it concise, in the line 59 unneceesarilly extra character please remove it. Do not use CCT acronym for cone calorimetry, it is known as cone calorimetry. In conclusion more emphasis should be on the success of the research and should indicate where these materials will find their use.

Finally, the reviewer appreciates the detailed research of these innovative research.

Response: First of all, thank you very much for your affirmation of the article. Then, we revised the article accordingly according to your comments.

After careful consideration, we shorted the introduction part and made it more concise. The unnecessary extra characters in the line 60 was removed. The abbreviation of CCT was changed to cone calorimetry.

Reviewer 3 Report

In this manuscript the authors studied the effect of different levels of BC and AHP on the flame-retardant properties of PLA-based composites. The manuscript contains relevant results and I recommend it for publication after some modifications, as in the following:

1 - What is the surface characteristic of the bamboo charcoal particles? These particles may agglomerate during mixing when hydrophobic plastic polymers, including PLA, are used. Please clarify in the Introduction.

2  - How to make sure the homogeneous mixture of the BC during preparation given no interface compatibilizers or coupling agents were added?

3 - It is not entirely sure why the DSC and FTIR are done. The effect of fillers on the thermal properties of composites was not added. This issue may improve the quality of the work. Some polymer/filler interaction proposal by FTIR results may help explain the better performance.

4 - An explanation as to why BC/PLA/AHP 20% presents lower strain at break than BC/PLA/AHP 15% would be good to add. Since the trend shows the increase in strain at break with increased concentration of filler.

5 - The authors should better clarify the origin of the improvement of the mechanical properties of the composites, especially when filler is used. Some papers working with natural fillers (DOI: 10.1016/j.eurpolymj.2018.11.031  and DOI: 10.1002/pc.25196 ) showed that a part of external stresses can be absorbed by the fillers, while some is dissipated by particle-particle and particle-polymer interactions. Moreover, other papers (DOI: 10.1007/s10924-019-01389-z and DOI: 10.1016/j.eurpolymj.2019.05.005) showed a correlation between amount of filler, dispersion and improved mechanical properties of composites. The authors are kindly invited to read and compare. Further discussion on this topic may improve the quality of the work.

Author Response

Point 1: What is the surface characteristic of the bamboo charcoal particles? These particles may agglomerate during mixing when hydrophobic plastic polymers, including PLA, are used. Please clarify in the Introduction. Finally the reviewer appreciates the detailed research of these innovative research.

Response 1: The surface and interior of BC have a large number of uniform fine hairs and pores, which are non-polar. Aggregation may occur in the mixing process of PLA with high BC content.  See added Lines 306-308.

Point 2: How to make sure the homogeneous mixture of the BC during preparation given no interface compatibilizers or coupling agents were added?

Response 2: By adjusting the processing parameters such as temperature, time and feeding speed, BC can be evenly mixed. However, if BC with a higher additive content (more than 55%) is needed, it cannot be evenly mixed without the use of interface compatibilizer or coupling agent.

Point 3: It is not entirely sure why the DSC and FTIR are done. The effect of fillers on the thermal properties of composites was not added. This issue may improve the quality of the work. Some polymer/filler interaction proposal by FTIR results may help explain the better performance.

Response 3: DSC analyses is performed in order to measure Tm, Tcc and Tc of pure PLA which are expected to show the crystallization performance of raw material-PLA. FTIR test is expected to observe the changes of groups before and after the preparation of the materials, indicating the state of raw materials used and the successful combination of fillers and PLA.

    In order to better explain the interaction between the filler and the polymer, FTIR curves of BC/PLA and BC/PLA/AHP composite materials were added in Fig.2d on page 8. The infrared results showed that BC was successfully introduced into the PLA matrix, and AHP and BC were embedded in the PLA matrix as seen from the SEM images in Fig.9 of fractured BC/PLA/AHP (25/50/25) samples.

Point 4: An explanation as to why BC/PLA/AHP 20% presents lower strain at break than BC/PLA/AHP 15% would be good to add. Since the trend shows the increase in strain at break with increased concentration of filler.

Response 4: Non-uniform pores and inclusions can be the defects leading to fracture. With the formation of sub-cracks (Ho 2014, Ref 17), the bending strain at the fracture decreases with the decrease in bending strength. When AHP was added at 15%, the elongation at break was the best, and the further addition of AHP increased the risk of non-uniformity in the preparation of composite materials, resulting in a significant decrease in the ductility of composites. See added Lines: 344-348.

Point 5: The authors should better clarify the origin of the improvement of the mechanical properties of the composites, especially when filler is used. Some papers working with natural fillers (DOI: 10.1016/j.eurpolymj.2018.11.031 and DOI: 10.1002/pc.25196) showed that a part of external stresses can be absorbed by the fillers, while some is dissipated by particle-particle and particle-polymer interactions. Moreover, other papers (DOI: 10.1007/s10924-019-01389-z and DOI: 10.1016/j.eurpolymj.2019.05.005) showed a correlation between amount of filler, dispersion and improved mechanical properties of composites. The authors are kindly invited to read and compare. Further discussion on this topic may improve the quality of the work.

Response 5: Thanks for your guidance. After studying the four literatures you provided, we further analyzed the mechanical properties and quoted them into the article to better clarify the origin of the improvement of the mechanical properties of composites. See added Lines: 320-324.

Round 2

Reviewer 3 Report

The paper is suitable for publication after the minor correction below.
Pge 9 line 321: "Scaffaro et al. (2018) [37] and Ferreira et al. (2019) [38] showed ..." instead "Ferreira et al. (2019) [37-38] worked with natural fillers, showed...". In fact, Scaffaro et al did not work with natural fillers.